# Identification of Antibody-Mediated Hydrolysis Sites of Oligopeptides Corresponding to the SARS-CoV-2 S-Protein by MALDI-TOF Mass Spectrometry

**DOI:** 10.3390/ijms241814342

**Published:** 2023-09-20

**Authors:** Anna M. Timofeeva, Sergey E. Sedykh, Pavel S. Dmitrenok, Georgy A. Nevinsky

**Affiliations:** 1SB RAS Institute of Chemical Biology and Fundamental Medicine, Novosibirsk 630090, Russia; 2Faculty of Natural Sciences, Novosibirsk State University, Novosibirsk 630090, Russia; 3Elyakov Pacific Institute of Bioorganic Chemistry of Far East Branch RAS, 100 let Vladivostoku Ave. 159, Vladivostok 690022, Russia

**Keywords:** SARS-CoV-2, COVID-19, autoimmunity, RBD, oligopeptide, coronavirus, IgG, catalytic antibodies, immunoglobulin G, MALDI-TOF spectrometry, Sputnik V vaccine

## Abstract

Antibodies recognizing RBD and the S-protein have been previously demonstrated to be formed in humans after SARS-CoV-2 infection and vaccination with the Sputnik V adenovirus vaccine. These antibodies were found to be active when hydrolyzing FITC-labeled oligopeptides corresponding to linear epitopes of the S-protein. The thin-layer chromatography method allows the relative accumulation of the reaction product to be estimated but cannot identify hydrolysis sites. This study used the MALDI-TOF MS method to establish oligopeptide hydrolysis sites. Using the MALDI-TOF MS method in combination with the analysis of known hydrolysis sites characteristic of canonical proteases allowed us to establish the unique hydrolysis sites inherent only to catalytically active antibodies. We have discovered two 12-mer oligopeptides to have six hydrolysis sites equally distributed throughout the oligopeptide. The other three oligopeptides were found to have two to three closely spaced hydrolysis sites. In contrast to trypsin and chymotrypsin proteases, the catalytically active antibodies of COVID-19 patients have their peptide bond hydrolyzed mainly after proline, threonine, glycine, or serine residues. Here, we propose a new high-throughput experimental method for analyzing the proteolytic activity of natural antibodies produced in viral pathology.

## 1. Introduction

SARS-CoV-2 is known to have four major structural proteins: S (spike), M (membrane), N (nucleocapsid), and E (envelope) [1,2]. The spike protein is responsible for viral entry by binding angiotensin-converting enzyme 2 (ACE2) on the host cell membrane [2,3]. Thus, antibodies capable of binding to the receptor-binding domain (RBD) and inhibiting the interaction of protein S with ACE2 are of interest for preventing and treating COVID-19. The S-protein remains the main target of many anti-SARS-CoV-2 drugs, including neutralizing monoclonal antibodies, vaccines, and other inhibitors. The further study of its structure will contribute to the development of more effective or broad-spectrum antiviral drugs. Given the critical role of the S-protein in coronavirus infection, the focus of our work was to analyze antibodies (S-IgG) specific to this protein.

B-cell receptors on lymphocytes can recognize individual epitopes rather than the whole antigen [4,5,6]. Therefore, epitope mapping is required for a better understanding of the mechanisms underlying antigen recognition by antibodies and for improving the design of therapeutic monoclonal antibodies and epitope vaccines [7,8,9]. Given that SARS-CoV-2, the virus that causes COVID-19 and has the potential to develop new pathogenic strains, was only recently discovered, the study of epitopes that stimulate antibody synthesis and T-cell response is still in progress [10,11,12]. The 12-mer oligopeptides have been synthesized and used to map S-protein epitopes of SARS-CoV-2 in [13]. Linear epitopes to be recognized by antibodies were identified using plasma samples from SARS-CoV-2-transfected patients. In our previous work, we analyzed the interaction of IgG from COVID-19-transfected and/or Sputnik-V-vaccinated patients with nine oligopeptides corresponding to the most recognized epitopes of the S-protein of SARS-CoV-2. Antibody fractions specific to the RBD and S-protein (RBD-IgG and S-IgG, respectively) were isolated. FITC-labeled oligopeptides were used, and hydrolysis products were separated by thin-layer chromatography (TLC) followed by fluorescence visualization. Thus, the accumulation of reaction products and characterization of the oligopeptide hydrolysis reaction by antibodies was carried out. Only six of the nine oligopeptides under study were shown to be hydrolyzed by antibodies from COVID-19 exposed and/or vaccinated patients [14]. It should be noted that IgG, which does possess an affinity for SARS-CoV-2 proteins, was not active in the hydrolysis of all the oligopeptides. Similar results regarding the hydrolysis of oligopeptides by antibodies from COVID-19 convalescents were recently described in [15].

It should be noted that catalytic antibodies hydrolyzing various substrates (RNA, DNA, and proteins) have been already reported for some diseases [16,17]. Antibodies hydrolyzing not only globular proteins but also their corresponding oligopeptides have been described. For example, antibodies specific to myelin basic protein (MBP) derived from the blood plasma of systemic lupus erythematosus [18] and multiple sclerosis [19] patients can hydrolyze four oligopeptides (17-, 19-, 21-, and 25-mer OP) corresponding to MBP epitopes. Antibodies against integrase from HIV-infected patients were reported to efficiently hydrolyze specific oligopeptides corresponding to recognizable epitopes of viral integrase [20,21]. Antibodies specifically hydrolyzing HIV-1 reverse transcriptase and integrase are potentially interesting for developing novel HIV vaccination strategies [22].

Biochemical characterization of the antibody-catalyzed hydrolysis reaction is performed using fluorescently labeled oligopeptides. The TLC separation of reaction products allows the accumulation of hydrolyzed oligopeptide fragments to be visualized. Analyzing the reaction product accumulation provides an opportunity to evaluate the oligopeptide hydrolysis efficiency by different antibody preparations and to characterize these reactions. However, this method does not allow identification of the hydrolysis sites. To solve the problem of establishing the sites of oligopeptide hydrolysis, we applied the MALDI-TOF MS method in this work. Separating the hydrolysis products of short oligopeptides with known molecular mass and amino acid sequences allows the peaks to be matched to the products formed. Notably, detecting hydrolysis products by MALDI-TOF MS does not require a fluorescent tag in the peptide. MALDI-TOF MS is used in the analysis of proteins and peptides. This method can be used to identify the changes in the structure of polypeptides, requires a very low number of samples, sample preparation is easy, and spectra are simple.

## 2. Results and Discussion

### 2.1. Characterization of Donor Groups and Antibody Preparations

The groups of 25 donors each involved in this study were described in our previous work [14]. These are COVID-19 survivors who were then vaccinated with Sputnik V (Con + Vac group), COVID-19 survivors (Con group), and healthy peoples vaccinated with Sputnik V (Vac group). The group of COVID-19-naïve and unvaccinated donors was not included in this study because the blood plasma antibody preparations of these donors did not show an affinity for the S-protein of SARS-CoV-2.

The S-protein of the SARS-CoV-2 virus is a major component of the viral envelope, responsible for virus entry and serving as a major target of host immune defense [23,24,25,26]. COVID-19 transfection results in the formation of antibodies to coronavirus proteins, including the S-protein [27]. Sputnik V vaccination also causes antibodies to the S-protein of the coronavirus to be formed [28,29].

Antibody fractions with an affinity for the S-protein referred to as S-IgG were obtained from the blood plasma of three groups of patients by affinity chromatography. It has been shown that S-IgG antibodies hydrolyze oligopeptides corresponding to the epitopes of the S-protein. Antibodies without affinity for the S-protein were not active in the hydrolysis of any oligopeptides that were used in this research [14].

In this work, we used five oligopeptides that are recognizable epitopes of the S-protein and are uniformly distributed over the protein surface to identify hydrolysis sites. Figure 1 presents a proposed structure of the SARS-CoV-2 S-protein, with the oligopeptides used in this work being highlighted in color.

### 2.2. MALDI Spectrometric Analysis of Oligopeptide Hydrolysis by Antibodies

This study analyzed the hydrolysis of five 12-mer oligopeptides by antibodies generated in response to the S-protein of the SARS-CoV-2 virus. These antibodies are polyclonal and are represented by fractions to the RBD and the remaining regions of the S-protein (RBD-IgG and S-IgG, respectively). The oligopeptides are the epitopes of the S-protein for which the formation of antibodies was detected [13]. MALDI-TOF spectrometric analysis of hydrolysis products of five oligopeptides (AR, TK, TQ, QG, and SN) by S-IgG antibody fractions was performed. Intact oligopeptides without antibodies in the reaction mixture were analyzed as a control.

Next, the results obtained for oligopeptide TQ are discussed. The data obtained for the remaining oligopeptides can be found in the Appendix A. Figure 2 shows the MALDI-TOF spectra of the antibody hydrolysis products of the TQ oligopeptide (TESNKKFLPFQQ).

In the region of less than 0.5 *m*/*z*, multiple peaks are observed, but they correspond to very short fragments of oligopeptides: less than 4 amino acid residues. We focused on larger fragments of oligopeptides, which correspond to individual, well-identified *m*/*z* peaks.

We calculated the monoisotopic masses of all theoretically possible fragments of the 12-linked oligopeptide (not shorter than peptides with a length of 4 amino acid residues). The MALDI-TOF spectrometry peaks obtained and the molecular masses calculated for possible fragments of the TQ oligopeptide were compared (Table 1). According to the data obtained, we determined the 12-mer oligopeptide fragments resulting from the hydrolysis of the oligopeptide by IgG: T↓ESN↓KK↓F↓LP↓FQ↓Q, with arrows denoting the sites of oligopeptide hydrolysis by antibodies. Thus, six sites of the TQ oligopeptide hydrolysis were detected.

We also performed MALDI-TOF spectrometric analysis for AR (Appendix A), TK (Appendix A), QG (Appendix A), and SN (Appendix A) oligopeptides. The MALDI-TOF spectrometry peaks obtained and the molecular masses calculated for the possible oligopeptide fragments were compared (Appendix A). The data obtained allowed us to determine the sites of hydrolysis of five 12-mer oligopeptides by antibodies. The results are summarized in Table 2.

The antibodies of Con + Vac donors hydrolyzed the TK oligopeptide at two T7↓E8 and S9↓N10 sites: TGTGVLT↓ES↓NKK. The antibodies of Con and Vac donors were characterized by another T1↓G2 hydrolysis site: T↓GTGVLT↓ES↓NKK. The resulting TQ, QG, and SN oligopeptide hydrolysis products were not different for antibodies from different donor groups. S-IgG hydrolyzed the TQ oligopeptide at 6 sites, QG at three sites, and SN at two sites. The hydrolyzed peptide bonds are visualized on 3D models of oligopeptides (Figure 3).

Figure 3 demonstrates that, for AR and TQ oligopeptides, the hydrolysis sites were located uniformly throughout the oligopeptide. Six hydrolysis sites were identified for each of these oligopeptides. Three oligopeptides (QG, TK, and SN) contained 2–3 closely located hydrolysis sites. The sites of hydrolysis were analyzed: Table 3 summarizes the major and minor hydrolysis sites and the matches of hydrolysis sites between the oligopeptides analyzed.

Table 3 demonstrates that antibodies hydrolyze the peptide bond most efficiently after threonine (T), proline (P), and lysine (K) residues. The peptide bond is less efficiently hydrolyzed after glycine (G), serine (S), and asparagine (N) residues. The hydrolysis sites after alanine (A) and glutamine (Q) residues have also been identified.

It should be noted that while preferential hydrolysis was detected after threonine (T), proline (P), and lysine (K) residues, hydrolysis at these bonds was not observed at all sites of the oligopeptide. This result is probably due to the peculiarities of amino acid recognition by antibodies and the conformational accessibility of the bond being hydrolyzed.

### 2.3. Comparison of the Specificity of the Hydrolysis of Five Oligopeptides by Antibodies Formed upon Vaccination and COVID-19 Infection and of the One by Proteases

Next, the sites of hydrolysis of the four 12-mer oligopeptides by proteases (trypsin and chymotrypsin) were analyzed and compared to the sites of hydrolysis by antibodies (Table 4).

Trypsin is known to hydrolyze proteins after positively charged lysine (K) and arginine (R) residues. In contrast, chymotrypsin hydrolyzes proteins preferentially after aromatic amino acids (tryptophan (W), phenylalanine (F), tyrosine (Y)) and leucine (L) [32,33,34].

The AR, TQ, and SN oligopeptides were found to have one hydrolysis site by antibodies after a lysine (K) residue. However, in contrast to trypsin that hydrolyzes the peptide bond after all lysine (K) residues, the antibodies were not detected to hydrolyze the peptide bond after all lysine (K) residues.

In SN and TQ peptides, the peptide bond hydrolysis after a phenylalanine (F) residue was detected, a characteristic feature of chymotrypsin [32,35,36]. However, the antibodies did not hydrolyze the peptide bond after phenylalanine (F) in all analyzed cases. The peptide bond hydrolysis by antibodies (in contrast to chymotrypsin) was not detected after tyrosine (Y) residues.

Thus, antibodies were found to hydrolyze predominantly the peptide bond after proline (P), neutral amino acids (threonine (T), glycine (G), serine (S)), and, in rare cases, after lysine (K) and phenylalanine (F) residues. Although some sites were found to be similar to those of trypsin and chymotrypsin, we did identify the unique sites characteristic only of antibody hydrolysis.

Proteolytic enzymes are known to exhibit proteolytic activity against any proteins and cleave the peptide bond after strictly defined amino acid residues (lysine and arginine in the case of trypsin and aromatic amino acids and leucine in the case of chymotrypsin). In contrast to classical proteases, antibodies are characterized by specific hydrolysis of the peptide bond. It is not just antibodies that exhibit selectivity for certain proteins. For example, in this study, a fraction of antibodies specific to the S-protein of SARS-CoV-2 has been isolated and shown to hydrolyze oligopeptides corresponding to this protein. Antibodies with catalytic activity also have selective specificity; this means that they do not hydrolyze the peptide bond after any amino acid residue that is typically susceptible to hydrolysis. This selectivity is probably associated with the recognition of all amino acid radicals in the specific sequences.

The present work demonstrates for the first time that catalytic antibodies are not simply capable of hydrolyzing peptide bonds. This type of proteolysis is not similar to the peptide bond hydrolysis by proteolytic enzymes such as trypsin and chymotrypsin. We have identified unique sites of oligopeptide hydrolysis that are specific only to antibodies and not to classical proteases: antibodies hydrolyze the peptide bond after proline (P), threonine (T), glycine (G), and serine (S) residues, which is not characteristic of canonical proteases.

Catalytic antibodies are a unique class of proteolytic enzymes capable of selectively hydrolyzing a peptide bond. Understanding the potential of antibodies may be of great promise for the development of monoclonal antibodies that specifically hydrolyze the peptide bond for application as part of an enzymatic system.

## 3. Materials and Methods

### 3.1. Donors and Patients

The study was approved by the Local Ethics Committee of the Institute of Chemical Biology and Fundamental Medicine (Protocol Number 21-4 from 15 August 2020), including the written consent of patients and healthy donors to present their blood for scientific purposes (according to the guidelines of the Helsinki ethics committee).

Vacuum tubes with anti-coagulation compound (EDTA) were used to collect fasting venous blood. Blood tubes were centrifuged at 3000× *g* for 15 min in a 5810 centrifuge (Eppendorf, Hamburg, Germany). The plasma separated from the red cell mass was divided into aliquots and stored at −70 °C.

The diagnosis of COVID-19 was made based on a PCR result at the acute stage of the disease and was subsequently confirmed by ELISA. SARS-CoV-2 IgG serology against the S-protein and N-protein of SARS-CoV-2 was evaluated using an Antigma G ELISA kit (Generium, Moscow, Russia) according to the manufacturer’s instructions. Plasma samples from 100 volunteers were selected for this study. Then 4 groups of donors of 25 each were formed: those who had COVID-19 and then were vaccinated with Sputnik V (Con + Vac group), those who had COVID-19 (Con group), those vaccinated with Sputnik V (Vac group), and a control group of COVID-19-naive and unvaccinated donors (Neg group). The Sputnik V vaccine, also known as Gam-COVID-Vac, contains adenovirus 26 (Ad26) and adenovirus 5 (Ad5) as vectors to express the recombinant gene corresponding to the S-protein of SARS-CoV-2.

Electrophoretically and immunologically homogeneous IgG from the blood plasma of patients was obtained by affinity chromatography on Protein-G-Sepharose (Cytiva, Uppsala, Sweden) in a similar way as described in [37]. The resulting IgG preparations were fractionated on sorbents with immobilized S-protein, yielding S-IgG fractions similar to [37,38]. The affinity chromatography profiles obtained in this work are similar to the results obtained in [38] and are not reported in this work. Additionally, the electrophoretic and immunologic homogeneity of the obtained preparations was checked similarly to [38].

### 3.2. Characterization of the Oligopeptides

The sequences of five fluorescently labeled 12-mer oligopeptides corresponding to different antibody-recognizable epitopes of the S-protein were selected and synthesized based on the data reported in the article [13] (Table 5). 

Oligopeptides were synthesized by Proteogenix (Schiligeheim, France). The quality control was performed by the manufacturer using MS and HPLC methods.

#### Visualization of the Oligopeptides

The location of oligopeptide sequences corresponding to the amino acid sequences of the SARS-CoV-2 S-protein was made in AlphaFold2 software [30] and RCSB PDB Mol* 3D Viewer [31].

### 3.3. MALDI-TOF Analysis of Oligopeptide Hydrolysis Sites

The reaction mixture (10 μL) contained 20 mM of Tris-HCl, pH 7.5, 10 mM of oligopeptide, and 0.01 mg/mL of IgGs. The reaction was started by adding antibodies, and the resulting mixture was incubated for 24 h at 37 °C.

The oligopeptide hydrolysis products were analyzed in all cases by MALDI-TOF mass spectrometry using a Reflex III system (Bruker, Bremen, Germany) with a 337 nm nitrogen laser (VSL-337 ND, Laser Science, Newton, MA, USA), with a pulse duration of 3 ns. A saturated solution of cyano-4-hydroxycinnamic acid in a mixture of 0.1% acetonitrile and trifluoroacetic acid (1:2) was used as a matrix (all reagents from Sigma, St. Louis, MO, USA). One μL of 0.2% trifluoroacetic acid and 2 μL of the matrix were added to 1 μL of the reaction mixture containing hydrolyzed oligopeptides before or after their separation by reversed-phase chromatography. One μL of the resulting mixture was applied to a MALDI plate, dried in the air, and used for the analysis. The MALDI spectra were calibrated using protein and OP I and II standards (Bruker Daltonic, Bremen, Germany) in external and internal calibration mode.

All possible oligopeptide sequences that could be formed by cleavage of a 12-mer oligopeptide were analyzed. The monoisotopic masses of all theoretical oligopeptide fragments were calculated using http://web.expasy.org/peptide_mass/ (accessed on 17 August 2023). The calculated mass and experimental results were compared, taking into account possible double- and triple-charged particles.

## 4. Conclusions

Catalytic antibodies can be regarded as a unique class of enzymes. These antibodies can recognize and hydrolyze specific peptide sequences. In this manuscript, we demonstrate that catalytic antibodies hydrolyze the peptide bonds after a number of amino acid residues different from those of classical proteases, such as trypsin and chymotrypsin. Unusual properties of proteolytic antibodies have been found: the hydrolysis of a peptide bond after proline (P), threonine (T), glycine (G), and serine (S) residues, which are not specific for canonical proteases.

Monoclonal catalytic antibodies may have the ability to hydrolyze the peptide bond at unique amino acid residues. Given that the repertoire of antibodies formed in response to vaccination may contain fractions of catalytically active antibodies that can not only specifically recognize the target protein but also hydrolyze it at specific sites, our results may be important for vaccine development. In this paper we have shown that the MALDI-TOF-MS method is convenient for the screening of catalytic antibodies generated at sites of hydrolysis using oligopeptides as substrates.

## Figures and Tables

**Figure 1 ijms-24-14342-f001:**
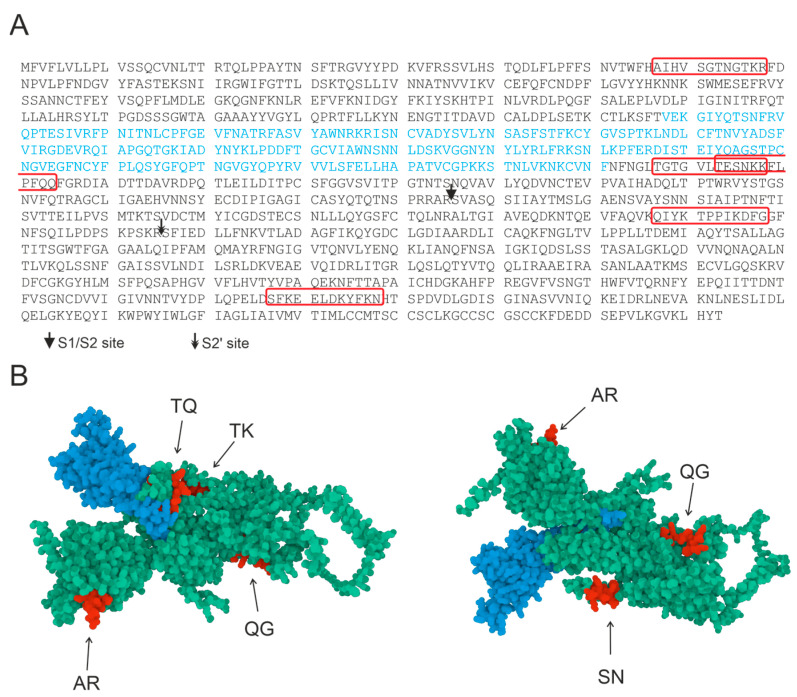
Amino acid sequence of SARS-CoV-2 S-protein (**A**). Three-dimensional model of S-protein (**B**). Sequence of RBD is marked in blue and sequences corresponding to the 12-mer OP sequences investigated in this work are marked in red. Visualization of OPs corresponding to the amino acid residues of the S-protein was performed in AlphaFold2 software (ColabFold v1.5.2-patch: AlphaFold2 using MMseqs2) [30] and RCSB PDB Mol* 3D Viewer [31].

**Figure 2 ijms-24-14342-f002:**
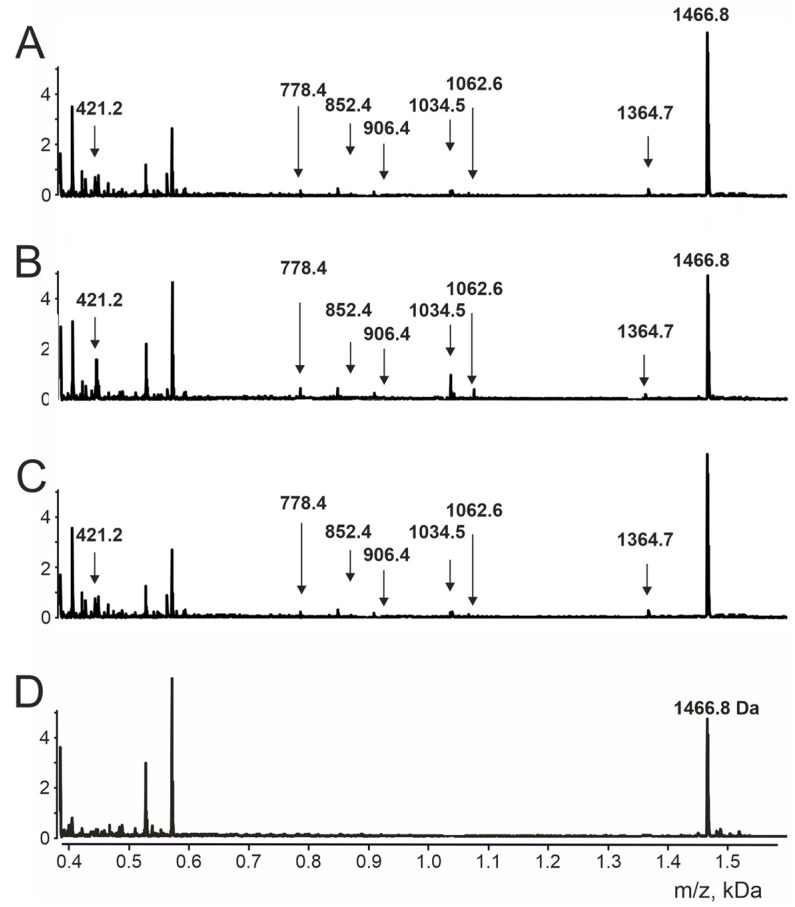
MALDI-TOF MS spectra of hydrolysis products of TQ oligopeptide by S-IgG antibodies isolated from Con + Vac group (**A**), Con group (**B**), and Vac group (**C**). (**D**) The MALDI-TOF MS spectra of intact oligopeptide. Peak 1466.8 corresponds to the intact oligopeptide TQ: TESNKKKFLPFQQ. The peaks with *m*/*z* less than 0.6 in the D spectrum correspond to matrix components.

**Figure 3 ijms-24-14342-f003:**
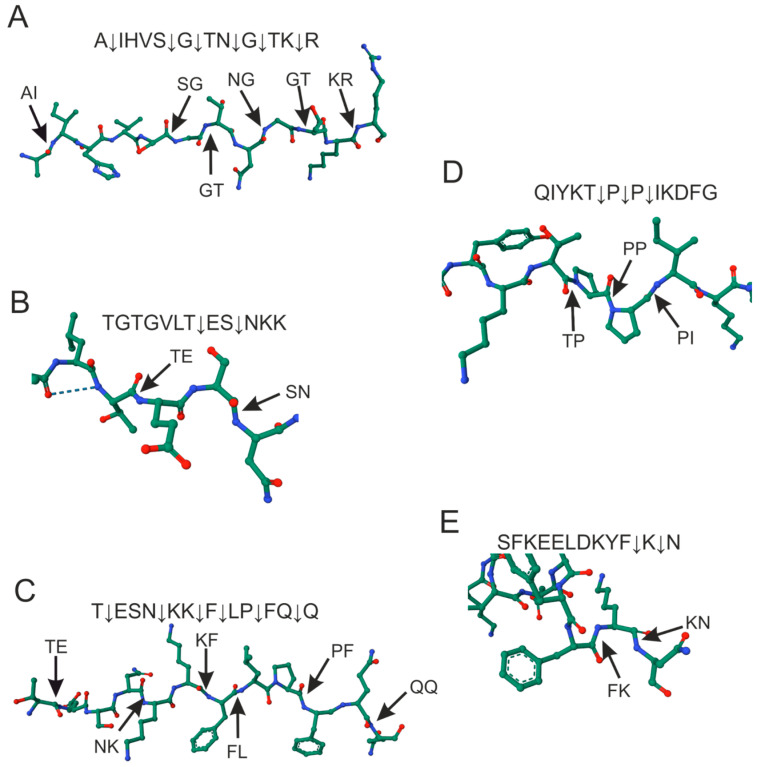
Hydrolysis sites of 12-mer AR (**A**), TK (**B**), TQ (**C**), QG (**D**), and SN (**E**) oligopeptides by RBD-IgG and S-IgG antibodies. Structures of OPs were visualized in AlphaFold2 software [30] and RCSB PDB Mol* 3D Viewer [31]. The arrows indicate the hydrolysis sites. The complete structures of AR and TQ oligopeptides are presented; in the case of other OPs only the hydrolysable fragment of the amino acid sequence is given. The letters correspond to the one-letter designations of amino acids.

**Table 1 ijms-24-14342-t001:** Comparison of the MALDI-TOF spectrometry peaks obtained and molecular masses calculated for possible fragments of the TQ oligopeptide.

Oligopeptide Fragment	Calculated Value kDa	Experimental Value *m*/*z*, kDa	Hydrolysis Site
FLPFQQ	778.9	778.4	K6↓F7
TESNKKF	852.9	852.4	F7↓L8
KKFLPFQ	907.1	906.4	N4↓K5, Q11↓Q12
KKFLPFQQ	1035.3	1034.5	N4↓K5
TESNKKFLP	1063.2	1062.6	P9↓F10
ESNKKFLPFQQ	1365.6	1364.7	T1↓E2

The arrows (↓) mean sites of hydrolysis

**Table 2 ijms-24-14342-t002:** The sites of hydrolysis of five oligopeptides by S-IgG antibodies from donors of three groups. The arrows indicate the hydrolysis sites.

Oligopeptide	Con + Vac	Con	Vac
AR	A↓IHVS↓G↓TN↓G↓TK↓R	A↓IHVS↓G↓TN↓G↓TK↓R	A↓IHVS↓G↓TN↓G↓TK↓R
TK	TGTGVLT↓ES↓NKK	T↓GTGVLT↓ES↓NKK	T↓GTGVLT↓ES↓NKK
TQ	T↓ESN↓KK↓F↓LP↓FQ↓Q	T↓ESN↓KK↓F↓LP↓FQ↓Q	T↓ESN↓KK↓F↓LP↓FQ↓Q
QG	QIYKT↓P↓P↓IKDFG	QIYKT↓P↓P↓IKDFG	QIYKT↓P↓P↓IKDFG
SN	SFKEELDKYF↓K↓N	SFKEELDKYF↓K↓N	SFKEELDKYF↓K↓N

**Table 3 ijms-24-14342-t003:** Analysis of peptide hydrolysis sites by antibodies. Amino acid residues after which hydrolysis occurs most frequently are indicated in green, amino acids after which hydrolysis rarely occurs are indicated in yellow, and amino acids after which hydrolysis is extremely rare are indicated in blue.

Oligopeptide	Amino Acids, after Which Hydrolysis by Antibodies Is Most Frequent (K, T, P)	Amino Acids, after Which Hydrolysis Is Rare (S, G, F)	Amino Acids, after Which Hydrolysis Is Extremely Rare (A, Q)
AR	A↓IHVS↓G↓TN↓G↓TK↓R	A↓IHVS↓G↓TN↓G↓TK↓R	A↓IHVS↓G↓TN↓G↓TK↓R
TK	T↓GTGVLT↓ES↓NKK	T↓GTGVLT↓ES↓NKK	T↓GTGVLT↓ES↓NKK
TQ	T↓ESN↓KK↓F↓LP↓FQ↓Q	T↓ESN↓KK↓F↓LP↓FQ↓Q	T↓ESN↓KK↓F↓LP↓FQ↓Q
QG	QIYKT↓P↓P↓IKDFG	QIYKT↓P↓P↓IKDFG	QIYKT↓P↓P↓IKDFG
SN	SFKEELDKYF↓K↓N	SFKEELDKYF↓K↓N	SFKEELDKYF↓K↓N

The arrows (↓) mean sites of hydrolysis.

**Table 4 ijms-24-14342-t004:** The sites of hydrolysis of four 12-mer oligopeptides by S-IgG antibodies and proteases. The hydrolysis site overlap between antibodies and proteases is indicated by color (between antibodies and trypsin is indicated by yellow, between antibodies and chymotrypsin—green).

	AR	TK	TQ	QG	SN
S-IgG	A↓IHVS↓G↓TN↓G↓TK↓R	T↓GTGVLT↓ES↓NKK	T↓ESN↓KK↓F↓LP↓FQ↓Q	QIYKT↓P↓P↓IKDFG	SFKEELDKYF↓K↓N
Trypsin	AIHVSGTNGTK↓R	TGTGVLTESNK↓K	TESNK↓K↓FLPFQQ	QIYK↓TPPIK↓DFG	SFK↓EELDK↓YFK↓N
Chymotrypsin	AIHVSGTNGTKR	TGTGVL↓TESNKK	TESNKKF↓LPF↓QQ	QIY↓KTPPIKDF↓G	SF↓KEEL↓DKY↓F↓KN

**Table 5 ijms-24-14342-t005:** Sequences of oligopeptides corresponding to the S-protein of SARS-CoV-2 used in the study.

12-mer OP	Position on S-Protein	Sequence
AR	67–78	AIHVSGTNGTKR
TK	547–558	TGTGVLTESNKK
TQ	553–564	TESNKKFLPFQQ
QG	787–798	QIYKTPPIKDFG
SN	1147–1158	SFKEELDKYFKN

## Data Availability

Most of the relevant raw experimental results are given in the Appendix A. Other empirical data that do not relate to the personal data of donors can be provided by request to Anna Timofeeva.

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
