# Peer review of "Identification of Antibody-Mediated Hydrolysis Sites of Oligopeptides Corresponding to the SARS-CoV-2 S-Protein by MALDI-TOF Mass Spectrometry"

_ijms, 2023, doi:10.3390/ijms241814342_

Round 1

Reviewer 1 Report

In the manuscript, authors Timofeeva et. al, have characterized the catalytic ability of anti SARS-CoV2-Spike antibodies. Authors have utilized the power of mass spectrometry for unambiguous assignment of digestion sites. My comments are below.

1)      Major flow of the study is the lack of control. Though authors have described in the material and method that they have isolated RBD-IgG and S-IgG but haven’t shown any data specific for RBD-IgG. RBD-IgG should only digest linear epitope in RBD-IgG and not in rest other S2-protein epitope peptides. Additionally, there should be a negative control such as antibodies coming in flow through of affinity chromatography and show that they have no effect on any of the peptides studied.

2)      It would be good to provide location of each peptide in linear sequence and indicate its position such as NTD, RBD, S2 region along with 3D structure for better understanding of sequence context of 12 mer peptides.

3)      Figure 2, panel A, B and C, there is a huge increase in mass close to 400 m/z but which is not there in plain peptide mass spectra. The authors haven’t talked about it.

4)      Table 2: It shows AR peptide is exclusively digested by “Con” group but not by “Con + Vac”. It should show digestion in “Con+Vac” group as well.

5)      Table 2: Digestion sites remain same across Con+Vac, Con, and Vac groups. This raises two questions. 1) is this observation indicating antibody pool generated by natural infection and Vaccinated individuals are very similar? 2) If yes, then what are the digestion efficiencies across groups?

6)      In figure S1 panel B, there is huge increase in mass about 400 m/z and 450 m/z but authors haven’t talked about it.

Author Response

1) Major flow of the study is the lack of control. Though authors have described in the material and method that they have isolated RBD-IgG and S-IgG but haven’t shown any data specific for RBD-IgG. RBD-IgG should only digest linear epitope in RBD-IgG and not in rest other S2-protein epitope peptides. Additionally, there should be a negative control such as antibodies coming in flow through of affinity chromatography and show that they have no effect on any of the peptides studied.

Previously, we characterized in detail the peptide hydrolyzing activity and showed that antibodies, which do not have an affinity for the S-protein, are not active in the hydrolysis of the OPs, corresponding to the epitopes of the S-protein. We added the characterization of this peptide hydrolyzing activity in lines 57–61.

This manuscript is devoted to the characterization of sites of hydrolysis of oligopeptides by the S-IgG subfractions. Therefore, this paper does not show the MALDI spectra of oligopeptides incubated with IgG which lack affinity for the S-protein (they correspond to the spectra shown in Fig. 2C). Currently, a manuscript devoted to other methods for analyzing peptide hydrolysis by antibodies in COVID-19 convalescents and vaccinated patients is being reviewed in the MDPI Vaccines.

2) It would be good to provide location of each peptide in linear sequence and indicate its position such as NTD, RBD, S2 region along with 3D structure for better understanding of sequence context of 12 mer peptides.

The location of the used oligopeptides on the linear sequence of the S-protein is shown in Fig. 1. The RBD region, sites S1/S2 and S2’ are indicated.

3) Figure 2, panel A, B and C, there is a huge increase in mass close to 400 m/z but which is not there in plain peptide mass spectra. The authors haven’t talked about it.

An increase in mass close to 400 m/z corresponds to very short oligopeptide fragments containing 2–4 amino acid residues. For calculations, we used peptide fragments not shorter than peptides length of 4 amino acid residues, shorter fragments are difficult to identify. We have added a description; please see lines 134–137.

4) Table 2: It shows AR peptide is exclusively digested by “Con” group but not by “Con + Vac”. It should show digestion in “Con+Vac” group as well.

There was a typo, which we have corrected. Please see in the Table 2.

5) Table 2: Digestion sites remain same across Con+Vac, Con, and Vac groups. This raises two questions. 1) is this observation indicating antibody pool generated by natural infection and Vaccinated individuals are very similar? 2) If yes, then what are the digestion efficiencies across groups?

  1. We have shown that antibodies hydrolyzing peptides corresponding to the S-protein of SARS-CoV-2 are formed both after COVID-19 and after vaccination. The point of vaccination is to imitate the immune response to a natural antigen, so the similarity of antibody pools generated as a result of vaccination and in convalescents is not surprising.
  2. Antibodies of the Con group more effectively hydrolyze the S-protein (data not yet published) and its oligopeptides (the article is under review in the MDPI Vaccines)

6) In figure S1 panel B, there is huge increase in mass about 400 m/z and 450 m/z but authors haven’t talked about it.

An increase in mass close to 400 m/z corresponds to the very short oligopeptide fragments consisting of 2-4 amino acid residues. For calculations, we used peptide fragments not shorter than peptides with a length of 4 amino acid residues since the shorter fragments are difficult to identify. We have added a description, please see lines 134-137.

Reviewer 2 Report

This research used MALDI-TOF Mass Spectrometry to identify the hydrolysis sites of the SARS-CoV-2 Sprotein. The research focuses on 5 12-mer oligopeptides. The research used the S-IgG of donnors, divided into 4 groups. In general, the hydrolysis sites were similar between the groups, with some minor differences on TK oligopeptide. The manuscript is well written, it is easy to read and to understand. It proves additional information regarding the host immune response against this virus. The vaccine was the Sputnik V. I wonder if similar results would be found using the other vaccines such as the mRNA vaccines.

Additional comments:

(1) Could you please add the catalog number of the Genarium Antigma G ELISA kit (line 230)?

(2) Lines 231-234. Regarding the 4 groups.

    (2.1) Could you please explain how the diagnosis of the patients who had had COVID-19 was assessed?

    (2.2) Could you please describe the characteristics/properties of the Sputnik vaccine?

(3) Figure 1 show the tridimensional structure of the S-protein. It is stated that two software were use to contruct this image: alphafold2 and rcsb pdb mol 3d viewer. Does the two software agree on the 3D image? What are the limitations of the AlphaFold models?

(4) Could you please add a 2D figure of the S-protein, showing the pathogenic function, and the target of host immune defense (line 86)?

(5) Line 32. Could you please add "receptor-binding domain" before "RBD"?

(6) Line 35. Could you please expand with "S protein remains the main target of many anti-coronavirus drugs, including neutralizing monoclonal antibodies (mAbs), vaccines, and other inhibitors. Further study of its structure will contribute to the development of more effective or broad-spectrum antiviral drugs"? (paraphrase it).

(7) Line 54. How antibodies hydrolize linear epitopes? Could you please describe the mechanism? Does the antibody use a passive catalytic strategy and does not chemically participate in the mechanism (non-canonical function)?

(8) Line 76. Could you please add that maldio-tof mass spectrometry is used in the analysis of proteins and peptides, and that changes in structure can also be identified. It requires very low amount of samples and sample preparation is easy and spectra is simple.

(9) In Figure 2. The maldi tof ms spectra was not different between the 3 groups (a, b, and c). Is this interpretation correct? Did you get the spectra of the negative group? Would the negative group whow the spectra of D?

Author Response

I wonder if similar results would be found using the other vaccines such as the mRNA vaccines.

In our country, most of the population has been vaccinated with the Sputnik V vaccine. Unfortunately, we do not have a sufficient collection of blood plasma from patients vaccinated with other vaccines. Since the protein synthesized during vaccination with Sputnik V and the analogs (adenoviral platform) and mRNA vaccines (Pfizer and Moderna) has in general, the same sequence, we expect that the results should be similar. In the case of inactivated whole-virion vaccines, we expect slightly different results. Unfortunately, the amount of blood plasma from patients vaccinated with Covivac (whole virion vaccine) in our collection is also not enough for publication. 

Дополнительные комментарии:

(1) Could you please add the catalog number of the Genarium Antigma G ELISA kit (line 230)?

We have clarified the name of the drug (please see line 256). Unfortunately, it does not have any catalog number. https://www.generium.ru/products/antigma/

(2) Lines 231-234. Regarding the 4 groups.

(2.1) Could you please explain how the diagnosis of the patients who had had COVID-19 was assessed?

The diagnosis of COVID-19 was made based on the PCR result at the acute stage of the disease and was subsequently confirmed by ELISA (the presence of IgG in the blood plasma). Please see lines 249-250.

(2.2) Could you please describe the characteristics/properties of the Sputnik vaccine?

We have added a description of the vaccine. See lines 257-259

(3) Figure 1 show the tridimensional structure of the S-protein. It is stated that two software were use to contruct this image: alphafold2 and rcsb pdb mol 3d viewer. Does the two software agree on the 3D image? What are the limitations of the AlphaFold models?

The purpose of Fig. 1 was to show the location of the oligopeptides used in the research on the surface of the S-protein. We used these tools not to build the model, but for visualization. We believe that AlphaFold's limitations are not significant in this case.

(4) Could you please add a 2D figure of the S-protein, showing the pathogenic function, and the target of host immune defense (line 86)?

We have given the location of the oligopeptides used on the linear sequence of the S protein. The RBD region, sites S1/S2 and S2’ were designated. See Fig.1

(5) Line 32. Could you please add "receptor-binding domain" before "RBD"?

We added the receptor-binding domain before the RBD, please see line 33

(6) Line 35. Could you please expand with "S protein remains the main target of many anti-coronavirus drugs, including neutralizing monoclonal antibodies (mAbs), vaccines, and other inhibitors. Further study of its structure will contribute to the development of more effective or broad-spectrum antiviral drugs"? (paraphrase it).

Please, see lines 35–37

(7) Line 54. How antibodies hydrolize linear epitopes? Could you please describe the mechanism? Does the antibody use a passive catalytic strategy and does not chemically participate in the mechanism (non-canonical function)?

The exact mechanism has not been described in the literature. Still, at least two mechanisms have been proposed: stabilization of the transition state (as in the case of active centers of enzymes), the production of such antibodies can be stimulated by stable analogs of transition states of a chemical reaction; the second mechanism is the production of anti-idiotypic antibodies against the active centers of enzymes (such antibodies are generated as a result of immunization with enzymes). Antibodies can exhibit a catalytic function (hydrolyzing) substrates such as DNA, RNA, proteins, oligopeptides, and other biomolecules. See, for example, a recent review from our laboratory [10.3390/ijms21155392].

(8) Line 76. Could you please add that maldio-tof mass spectrometry is used in the analysis of proteins and peptides, and that changes in structure can also be identified. It requires very low amount of samples and sample preparation is easy and spectra is simple.

We added these sentences, please see lines 84–87.

(9) In Figure 2. The maldi tof ms spectra was not different between the 3 groups (a, b, and c). Is this interpretation correct?

The obtained spectra did not differ practically between the groups, except for the TK oligopeptide, which did not have a hydrolysis site at positions 1-2. Previously, we characterized in detail the peptide-hydrolyzing activity and showed that antibodies that do not have an affinity for the S-protein are not active in the hydrolysis of the oligopeptides corresponding to the epitopes of the S-protein.

Did you get the spectra of the negative group? Would the negative group show the spectra of D?

In this work, we do not present MALDI spectra of oligopeptides incubated with IgG that do not possess any affinity for the S-protein. Your guess is correct, they correspond to the spectrum shown in Fig. 2D.

Round 2

Reviewer 1 Report

Authors have addressed all my comments satisfacotry and hence, I am recommending it for publication. 

I noticed one typo on line 132. it should be 0.5 k*m/z and not only 0.5 m/z. It is minor point but need to be corrected.